# Study on energy rebound effects of China's industries

**Guangqing Xu[1], Danze Huang[2], Xiaoyu Chen[3], Mengyao Chen[1] \***

**1** School of Environment and Natural Resources, Renmin University of China, Beijing, China, **2** School of Science, Hong Kong University of Science and Technology, Hong Kong, China, **3** Department of Strategic Investment, Jinshan Capital Group, Shanghai, China

\* 18811552272@163.com

## Abstract

From the perspective of generalized technological progress, this study investigates the energy rebound effects on China's domestic overall and the country's various industries using 2005–2017 data. The results showed that the rebound effect of the domestic overall is driven mainly by the secondary sector. The domestic overall and high-energy-consuming industries decrease the rebound effects, whereas low-energy-consuming industries increase the rebound effects. As China's industrial structure does not lean toward high-energy-consuming industries, this implies that the Industrial Restructuring Initiative that began in the Chinese government's "Eleventh Five-Year Plan" has effectively slowed down the rebound effects of high-energy-consuming industries, and the feedback effect of self-reinforcement within the economic system was offset. The results also implied that energy efficiency policies should shift to low-energy-consuming industries; the rebound effects of the energy industries could be lessened, the energy efficiency of low-energy-consuming industries could be increased, and the industrial structure could be further optimized by implementing an appropriate pricing in electricity.

## I. Introduction

In order to cope with climate change, China has promised to achieve carbon peak and carbon neutrality before 2030 and 2060 respectively. There are amount of studies having predicted the carbon emissions of the economy overall or some industries in China under different scenarios [1–6].

However, the above studies have not taken into account the impact of energy rebound effect when predicting future carbon emissions. A study indicates an "energy rebound effect" represented by the phenomenon that the actual energy savings resulting from energy efficiency improvements are less than anticipated [7]. China's annual energy consumption per unit of gross domestic product (GDP) has reduced in recent years, and improvements in energy efficiency have been remarkable owing to technological progress and a series of energy efficiency improvement policies and measures implemented by the Chinese government. In this way, under the influence of energy rebound effect, the final forecast result may deviate from the

**Data Availability Statement:** All the data are obtained from the China Statistical Yearbook, the China Energy Statistical Yearbook and the China Industry Statistical Yearbook over the years. The data underlying the results presented in the study are available from the National Bureau of Statistics

of the People's Republic of China(http://www.stats.gov.cn/tjsj./). Anyone would be able to access or request these data in the same manner as the authors and the authors did not have any special access or request privileges that others would not have.

**Funding:** This work was only supported by the State Grid Corporation of China Headquarters Management Technology Project (Project No. 1400-202224242A-1-1-ZN). The funder had no role in study design, data collection and analysis, decision to publish, or preparation of the manuscript. There was no additional external funding received for this study.

**Competing interests:** The authors have declared that no competing interests exist.

actual situation in the future, thus affecting the realization of China's carbon emission reduction target. When predicting carbon emissions based on energy consumption, the energy rebound effect should be taken into account by adding a discount rate into the energy efficiency improvement related energy reduction, which may be represented as: *Actual Energy Savings = Theoretical Energy Savings* \* (1−*RE*), Where, *RE* means energy rebound effect.Therefore, it is very important to take the energy rebound effect into account in the in depth carbon emission reduction scenario analysis and corresponding policy makings.

By calculating the energy rebound effects of various industries at all levels during 2005–2017, this study examines whether the status quo in China exhibits industrial heterogeneity and the underlying reasons, and whether such a breakdown of the overall economy contributes to the current study in this regard. By analyzing the research results, several conclusions are drawn and some valuable policy suggestions are put forward, with a view to providing theoretical basis and reference for future industrial structure adjustment in different regions, improving energy efficiency and reducing energy consumption through technological innovation, and providing some valuable policy suggestions for China's later energy policy and carbon reduction policy, which is of theoretical and practical significance.

## II. Literature review

The notion of "energy rebound effect" can be traced back to as early as 1865 when British economist Jevons first proposed that energy efficiency improvements are often accompanied by broader technological progress, which accelerates the entire social reform and economic growth and, in turn, increases energy demand. Meanwhile, energy efficiency improvements reduce energy costs. The combined effect causes higher energy consumption than anticipated, and the result is named the "Jevons Paradox." Later on, Saunders proved the existence of the rebound effect in the framework of neoclassical growth theory, and categorized it into five levels: super-conservation, zero rebound, partial rebound, full rebound, and backfire [8]. There has been growing concern over energy issues in recent years, and studies on the energy rebound effect gain depth by degrees.

There are two main categories of methodology on the study of energy rebound effect on macro-integration, interprovincial regions and varies industries, the one is based mainly on the economic growth theory, and the other is based on the computable general equilibrium (CGE) model and energy input–output table. The former derives the estimation formula of rebound effects from logical relationships among energy efficiency or technological progress, economic growth, energy consumption, and the energy rebound effect; from a systematic and structural perspective, the latter estimates the output elasticity of energy in different industries using the energy input–output table, specifically, it assumes an energy efficiency improvement rate, and then simulates the energy rebound effect of the overall economy and different industries.

When using the economic growth method, a key variable is energy efficiency. Several means were used to measure the improvement of energy efficiency. First, construct an energy efficiency index by available data and estimate the energy rebound effect by calculating the elasticity of energy demand to energy efficiency index: By using this kind of method, Lin and Du [9] estimated that the rebound effect of China's overall economy was 30%-40%, with an average of 34.3% during 1981–2011. Li et al. [10] estimated the rebound effect of China in 2013 was 20.4% by using provincial level panel data. Yan et al. [11] estimated the average short-term and long-term direct rebound effects of Chinese provinces at 88.55% and 77.50% respectively. Chen et al. [12] estimated that the energy rebound effect in China was 79.94% on average from the macro level.

Second, it is assumed that energy price are exogenous to energy efficiency and changes in energy price and energy efficiency are symmetrical, and the energy rebound effect is estimated by calculating the elasticity of energy demand to energy price: By using this kind of method, Lin and Li [13] estimated the direct rebound effect of China's heavy industrywas about 74.3%. Zhang et al. [14] estimated that the direct rebound effect of national road passenger transport was 25.53% and 26.56% in the short and long term respectively. Liu et al. [15] divided the direct rebound effect into substitution and output channels, and found that the direct rebound effect of industrial sectors was 37.0%, and the contribution of substitution and output channels was 13.1% and 23.9% respectively. Fei et al. [16] studied that the energy rebound effect of China's agricultural sector was 74.78%.

Third, since China's energy price data is difficult to obtain, also it cannot accurately reflect the market reality, scholars used the decline in energy intensity to replace the improvement of energy efficiency. It is assumed that energy efficiency improvement is equivalent to technological progress, and the energy rebound effect is estimated by calculating the economic growth brought by technological progress and the increase of energy consumption: Zhou and Lin [17] were the first scholars who used the C-D production function to estimate that China's overall energy rebound fluctuated between 30% and 80%. In recent years, by using this kind of method, Wu et al. [18] estimated the national average carbon rebound is between 7.4% and 43.8%. Shao et al. [19] estimated the energy rebound effect of Shanghai, and found that the average energy rebound effect of Shanghai's overall economy and secondary industry was 93.96% and 73.10% respectively.

A few literatures using CGE model. Zha and Zhou [20] simulated the impact of a 4% rise in energy efficiency in the Chinese economy based on the CGE model and found that the rebound effects of coal, oil, and electricity were all close to 33%. Hu [21] ran CGE-based scenario simulations to compare the performance of the reduction in China's overall energy consumption and overall energy intensity in case of a 10% rise in energy efficiency in high-energy-consuming and low-energy-consuming industriesrespectively. Liu et al. [22] studied the energy conservation effect and rebound difference of various energy efficiency improvements by building a static CGE model that introduced the rebound effect calculation module. Feng [23] calculated the energy rebound effect of China's industries and the overall economy by compiling a value-based energy input–output table, with comparable energy physical flows. Du et al. [24] found that when the efficiency of the four major transportation sectors was increased by 10% each, the rebound effect of the whole economy was 9.46%, and the rebound effect of the whole transportation sector was 43.76% by using the CGE model.

In addition, some scholars studied the influencing factors of rebound effect. Wang and Nie [25] established a Cournot competition model and studied the influence of competition degree and price fluctuation on energy rebound effect. Miao and Chen [26] developed a non-parametric framework variable-specifically and source-specifically to investigate the impact mechanism of endogenous Total Factor Productivity (TFP) on rebound effect (RE).

However, the aforementioned study, which starts from a systematic and structural perspective, has certain imperfections, namely, overlooking energy prices owing to data limitations, ignoring direct rebounds within industries limited by functional settings, and the complex calculation of the CGE model, high-caliber demand of data samples, and strong sensitivity of the results to the settings of initial parameter values and production function forms. Based on the correlations of various industries in the economic system, studies based on the CGE model and energy input–output table are suited for the analysis of the generation mechanism of energy rebound effects and mutual influence among industries. However, many studies in this category focus on the horizontal correlation of industries in a certain year, without comparing the dynamic changes over time.

Nevertheless, literature on China's rebound effects focuses on macro-integration, interprovincial regions, and other study objects, with few studies on changes in energy rebound effects of various industries, existing problems, and reasons behind a perspective of industrial structure adjustment. With reference to the conclusions of the studies that have adopted a systematic and structural perspective, this study addresses the lack of dynamic analysis over time, estimates the energy rebound effects of domestic overall and various industries through empirical evidence and by applying the DEA–Malmquist index method based on the generalized technological progress, and analyzes the internal mechanism and external environment parameters that affect the energy rebound effects of various industries.Moreover, this study has two innovations. First, while building a theoretical model of the industrial energy rebound effect, the industrial rebound effect is innovatively categorized into three impact parameters, which are further simplified into two main parameters: the growth rate of TFP (GTFP) and the decrease rate of energy intensity (DEI), simplifying the analysis of the influencing parameters of industrial rebound effects. Compared with the traditional calculations of energy rebound effects, which is based on the ratio of new energy increment to theoretical energy savings, the decomposition of industrial energy rebound effect in this study directly analyzes the practical factors affecting its change. Second, the domestic overall economy is grouped by four levels from the perspective of industrial restructuring, including domestic overall, and primary, secondary, tertiary sectors. The secondary sector is divided into industry and construction. Industry is divided into energy industries: high-energy-consuming industries and low-energy-consuming industries. High- and low-energy-consuming industries are further classified into 13 subsectors.

## III. Model method and data source

### (I) Energy rebound estimation formula based on the generalized technological progress

Saunders [27] and Berkout et al. [28] defined the energy rebound effect as the ratio of the difference between the theoretical and actual energy savings to the theoretical energy savings after energy efficiency improvements, such difference called the energy rebounded.

$$RE = \frac{Energy\ Rebounded}{Theoretical\ Energy\ Savings} \tag{1}$$

Where,

$$Energy\ Rebounded = Theoretical\ Energy\ Savings - Actual\ Energy\ Savings \tag{2}$$

Thereby,

$$RE = 1 - \frac{Actual\ Energy\ Savings}{Theoretical\ Energy\ Savings} \tag{3}$$

$$Actual\ Energy\ Savings = Theoretical\ Energy\ Savings * (1 - RE) \tag{4}$$

Since the *Actual Energy Savings* is hard to measure, we use *Energy Rebounded* and *Theoretical Energy Savings* to estimate the rebound effect.

With reference to Zhou and Lin [17], suppose $Y_t$ refers to the actual yield, $E_t$ refers to the energy consumption, and $EI_t$ refers to the energy intensity; then, the formula of energy consumption in year $t$ is expressed as: $E_t = Y_t \cdot EI_t$. In year $t+1$, the economic yield is $Y_{t+1}$, the energy intensity decreased from $E_t$ to $EI_{t+1}$ because of technological progress, then, the formula

of energy consumption in year $t+1$ is expressed as: $E_{t+1} = Y_{t+1} \cdot EI_{t+1}$. Suppose there is no technological progress, the energy consumption in year $t+1$ will be: $E_{t+1} = Y_{t+1} \times EI_t$. Thereby we get the energy savings because of energy intensity decrease: $Y_{t+1} \cdot (EI_t - EI_{t+1})$. We call it *Theoretical Energy Savings*, it means the anticipanted energy savings from technological progress.

$$Theoretical\ Energy\ Savings = Y_{t+1} \cdot (EI_t - EI_{t+1}) \qquad (5)$$

However, technological progress promotes the economic yield, let $\sigma_{t+1} \cdot (Y_{t+1} - Y_t)$ be the economic growth brought by technological progress, where $\sigma_{t+1}$ is the contribution rate of technological progress to economic growth. On the other hand, economic growth promoted by technological progress drives more energy consumption, the increase in energy consumption is obtained by multiplying the economic growth brought by technological progress by the energy intensity in year $t+1$. Therefore, the increase in energy consumption in year $t + 1$, that is, *Energy Rebounded*, is expressed as:

$$Energy\ Rebounded = \sigma_{t+1} \cdot (Y_{t+1} - Y_t) \cdot EI_{t+1} \qquad (6)$$

From the above derivation, the *Energy Rebounded* is in fact the increase in energy consumption due to economic growth brought by technological progress. The energy rebound effect in year $t+1$ could be expressed as:

$$RE_{t+1} = \frac{\sigma_{t+1} \cdot (Y_{t+1} - Y_t) \cdot EI_{t+1}}{Y_{t+1} \cdot (EI_t - EI_{t+1})} \qquad (7)$$

However the yield promoted by the technological progress in a narrow sense is hard to measure, with reference to the existing literature [17], we define the technological progress in its generalized sense, which includes not only technological innovation but also improvements in the management efficiency and institutional factors. Generally, generalized technological progress can be measured by the Total Factor Productivity (TFP).

As both the actual yield $Y$ and energy intensity $EI$ can be obtained or calculated from the statistical yearbook, the key of calculating the energy rebound effect is to determine the contribution rate of technological progress to economic growth rate $\sigma_{t+1}$.

## (II) Estimation of $\sigma_{t+1}$

First, suppose the production function including three input factors of capital $K$, labor $L$ and energy $E$ as follows:

$$Y = TFP(t) \cdot F(K, L, E) \qquad (8)$$

where $Y$ represents total yield, $TFP(t)$ is total factor productivity, which represents generalized technological progress that change over time, $K$ represents capital, $L$ represents labor, and $E$ represents energy, that is, the total yield of the economy is determined by these four factors. Then, take the derivative of $t$ on both sides of the equation to get:

$$\frac{dY}{dt} = \frac{dTFP}{dt} \cdot F(K, L, E) + TFP \cdot \frac{dF}{dK} \cdot \frac{dK}{dt} + TFP \cdot \frac{dF}{dL} \cdot \frac{dL}{dt} + TFP \cdot \frac{dF}{dE} \cdot \frac{dE}{dt} \qquad (9)$$

Divide both sides of the equation by $Y$ to get:

$$\frac{dY}{dt \cdot Y} = \frac{dTFP}{dt \cdot TFP} + TFP \cdot \frac{dF}{dK} \cdot \frac{dK}{dt \cdot Y} + TFP \cdot \frac{dF}{dL} \cdot \frac{dL}{dt \cdot Y} + TFP \cdot \frac{dF}{dE} \cdot \frac{dE}{dt \cdot Y} \qquad (10)$$

Suppose that in a perfectly competitive market, the price of an input factor is equal to its marginal product, so the following equation exits, where $r$, $w$, $p$ represent the price of capital,

labor and energy respectively, and $\alpha$, $\beta$, $\gamma$ represent the output elasticity of capital, labor and energy respectively.

$$\begin{cases} r = TFP \cdot \dfrac{dF}{dK} = \alpha \cdot \dfrac{Y}{K} \\[2mm] w = TFP \cdot \dfrac{dF}{dL} = \beta \cdot \dfrac{Y}{L} \\[2mm] p = TFP \cdot \dfrac{dF}{dE} = \gamma \cdot \dfrac{Y}{E} \end{cases} \tag{11}$$

Thus:

$$\frac{dY}{dt \cdot Y} = \frac{dTFP}{dt \cdot TFP} + \alpha \cdot \frac{dK}{dt \cdot K} + \beta \cdot \frac{dL}{dt \cdot L} + \gamma \cdot \frac{dE}{dt \cdot E} \tag{12}$$

Eq (12) can be written as:

$$\frac{\Delta Y}{Y} = \frac{\Delta TFP}{TFP} + \alpha \cdot \frac{\Delta K}{K} + \beta \cdot \frac{\Delta L}{L} + \gamma \cdot \frac{\Delta E}{E} \tag{13}$$

Then, the growth rate of economic yield can be obtained as follows:

$$GY = GTFP - \alpha \cdot GK - \beta \cdot GL - \gamma \cdot GE \tag{14}$$

Where, $GY$, $GTFP$, $GK$, $GL$ and $GE$ respectively represent the growth rate of $Y$, $TFP$, $K$, $L$ and $E$ relative to the previous year. According to Eq (14), the contribution rate of the growth rate of $TFP$ and the growth rate of each input factor to the growth rate of $Y$ can be estimated respectively. Among them, the contribution rate of the growth rate of $TFP$ to the growth rate of $Y$ is as follows:

$$\sigma_{t+1} = \frac{GTFP_{t+1}}{GY_{t+1}} \tag{15}$$

## (III) Determining the GTFP via the DEA–Malmquist index method

In this study, the growth rate of $TFP$ that characterizes generalized technological progress is estimated by the DEA–Malmquist index method. DEA is a quantitative method that evaluates the efficiency of study objects using linear programming [29]. Essentially, the $TFP$ refers to the output efficiency that considers various input factors in the economic system, thereby applying the DEA method to estimate the $TFP$ [30].

This study estimates the $GTFP$, whereas the DEA model obtains the relative efficiency. The DEA–Malmquist index method builds the production frontiers of the two periods and calculates the corresponding TFP of the study objects successively, and the growth rate between the two TFPs refers to the generalized technological progress rate.

According to the method to calculate the Malmquist productivity index adopted by Fare R et al. [31], the geometric average of the TFP index of the frontiers with reference sets of periods 1 and 2 is taken as the Malmquist index of the study object. In general cases, the Malmquist index from period $t$ to period $t + 1$ can be expressed as

$$M(x^{t+1},\ y^{t+1},\ x^t,\ y^t) = \sqrt{\frac{R^t(x^{t+1},\ y^{t+1})}{R^t(x^t,\ y^t)} * \frac{R^{t+1}(x^{t+1},\ y^{t+1})}{R^{t+1}(x^t,\ y^t)}} \tag{16}$$

where $R^t$ represents the frontier with a reference set of period $t$, $R^{t+1}$ represents the frontier with a reference set of period $t+1$, and the frontiers are structured using linear programming.

$X^t$ and $x^{t+1}$ represent the inputs of the TFP in period $t$ and period $t + 1$, respectively, including capital, labor, and energy, $y^t$ and $y^{t+1}$ represent the output index of TFP in period $t$ and period $t + 1$, respectively, that is, economic yield.

As the DEA–Malmquist model involves intertemporal calculations, if a certain yield of the evaluated decision-making unit ($DMU_k$) is higher than that of other decision-making units that form the frontier, $DMU_k$ will not find a suitable reference object, and there will be no feasible solutions. At present, no studies have analyzed the possible case of a zero-feasible solution by applying this method. In this study, two methods are adopted to address the lack of feasible solutions in calculating the DEA–Malmquist index method: one is the frontier alternative index method and the other is the DEA–Malmquist model using the adjacent joint frontier reference [32, 33].

The changes in the TFP of the economy as a whole and of various industries in the two adjacent periods are represented by the Malmquist index obtained from Eq (6), as follows:

$$M(x^{t+1},\ y^{t+1},\ x^t,\ y^t) = \frac{TFP_{t+1}}{TFP_t} \tag{17}$$

When $M > (=, <)1$, it represents that the TFP increases (remains unchanged, declines), respectively. This study aims to obtain the ratio of GTFP to the growth rate of the economic yield, as follows:

$$\sigma_{t+1} = \frac{GTFP_{t+1}}{GY_{t+1}} \tag{18}$$

where $GTFP$ represents the growth rate of the TFP, that is, the growth rate of technological progress, and $GY$ (growth rate of $Y$) represents the growth rate of the economic yield, and we obtain the following:

$$GTFP_{t+1} = \frac{TFP_{t+1} - TFP_t}{TFP_t} = \frac{TFP_{t+1}}{TFP_t} - 1 = M(x^{t+1},\ y^{t+1},\ x^t,\ y^t) - 1 \tag{19}$$

$$GY_{t+1} = \frac{Y_{t+1} - Y_t}{Y_t} \tag{20}$$

Thus, we obtain the contribution rate of technological progress in year $t + 1$ as follows:

$$\sigma_{t+1} = \frac{M(x^{t+1},\ y^{t+1},\ x^t,\ y^t) - 1}{\frac{Y_{t+1} - Y_t}{Y_t}} \tag{21}$$

By substituting Eq (21) into Eq (7), the formula of the energy rebound effect can be obtained:

$$RE_{t+1} = GTFP_{t+1} \times \frac{1}{GY_{t+1} + 1} \times \frac{1}{DEI_{t+1}} \tag{22}$$

where $DEI$ represents the rate of decrease in energy intensity:

$$DEI_{t+1} = \frac{EI_t - EI_{t+1}}{EI_{t+1}} \tag{23}$$

Eq (22) shows that the energy rebound effect is affected mainly by three parameters: (1) the greater the $GTFP$, the greater the energy rebound effect; (2) the lower the growth rate of the economy, the greater the energy rebound effect; and (3) the lower the $DEI$, the greater the energy rebound effect. Thus, from the three parameters, it is concluded that the impact of the

growth rate of the economy on the energy rebound effect is smaller than that of the other two parameters, and the economic growth rate affects the *DEI*. Therefore, the three parameters affecting the energy rebound effect can be divided into *TFP* and energy intensity.

Characterizing the energy efficiency by the reciprocal of energy intensity is a nonstrict definition of the energy rebound effect, which combines information on economic yield and energy conservation, and is easier to estimate [34]. However, the decline in energy intensity not only characterizes the improvements in the energy efficiency resulting from technological progress but is also affected by the size of the added value of the industry's products. The size of added value reflects not only the industry's level of technological development but also the result of the industry's products being affected by supply and demand relationships or pricing control. Essentially, characterizing the energy efficiency by the reciprocal of energy intensity equals characterizing the energy efficiency in an economic sense [23].

Considering the fluctuations in the energy intensity of various industries in adjacent years, this study calculates the energy intensity for 2-year periods, divides the study period 2005–2017 into six periods, and calculates the energy rebound effect resulting from the energy efficiency improvements in 2017 relative to 2005 to improve the accuracy of the calculation.

Eq (13) shows that two extreme cases are likely to occur, namely, the GTFP, which characterizes technological progress, may be negative, and the energy intensity rises instead of declining. Both are not consistent with the definition of rebound effect and should be eliminated.

## (IV) Data source and processing

The aforementioned model shows that the data needed to estimate the energy rebound effect of China's industries include economic yield (Y), capital stock (K), labor (L), and energy consumption (E). The specific data source and processing are as follows:

1. Yield data Y
   In this study, the added value of domestic overall and various industries during 2005–2017 is chosen as yield data, with 2005 as the base period. The data are derived from the *China Statistical Yearbook* over the years (constant price in 2005).

2. Capital stock K
   Currently, the perpetual inventory method is applied widely in China and abroad to estimate the capital stock of various industries. In this study, the capital stock in 2005 compiled by TIAN Youchun is adopted as that of the base period, and the new investment in industries is measured by the fixed asset sequence of the whole society; and the different depreciation rates by Different industries proposed by Wu [35] is adopted for calculation. Data in different years are deflated using the fixed asset investment price index. The data are derived from the *China Statistical Yearbook* over the years.

3. Labor data L
   In this study, the employment figures across industries in China during 2005–2017 are chosen as the labor input, with a unit of 10,000 people. The labor data are obtained from the *China Statistical Yearbook* and the *China Industry Statistical Yearbook* over the years.

4. Energy consumption data E
   In this study, four types of energy sources, coal, natural gas, oil, and electricity, are converted into the standard coal equivalent for calculation. In addition, power is generated by converting the coal consumption into electricity. The end-use energy consumption of various industries in China during 2005–2017 is chosen as the energy input. The data are derived from the *China Energy Statistical Yearbook* over the years.

## (V) Descriptive statistics of model variables

In this study, the calculation of the energy rebound effect covers China's overall economy and primary, secondary, and tertiary sectors. The secondary sector includes industry and construction; in turn, the industry sector includes the energy industry, further categorized as high- and low-energy-consuming industries, which are further classified into 13 subsectors at four levels (Fig 1). As the study focuses on the rebound effects, the secondary sector features the highest proportion and the strongest energy intensity in China's total energy consumption. In this study, the energy industry and the energy-consuming industry are discussed separately. Specifically, various energy industries that produce and provide energy for all sectors of society are classified jointly as the energy industry, including coal mining and washing, oil and natural gas mining, petroleum processing, coking and nuclear fuel processing, electricity, heat and gas production, and supply. In terms of high-energy-consuming industries in the industrial sector, according to the *Statistical Bulletin of the People's Republic of China on National Economic and Social Development* (2017), which defines six high-energy-consuming industries and excludes petroleum processing, coking and nuclear-fuel-processing industries have been classified as an energy industry, as well as electricity, heat and gas production, and supply. High-energy-consuming industries include manufacturers of chemical raw materials and chemical products, manufacturers of nonmetallic mineral products, and smelting and rolling processing of metals (including ferrous and nonferrous metals); whereas the remaining industries are classified as the low-energy-consuming industry.

This study focuses on the first three levels of the analysis of the energy rebound effect. Table 1 shows the descriptive statistics of six industries, including the primary sector, energy

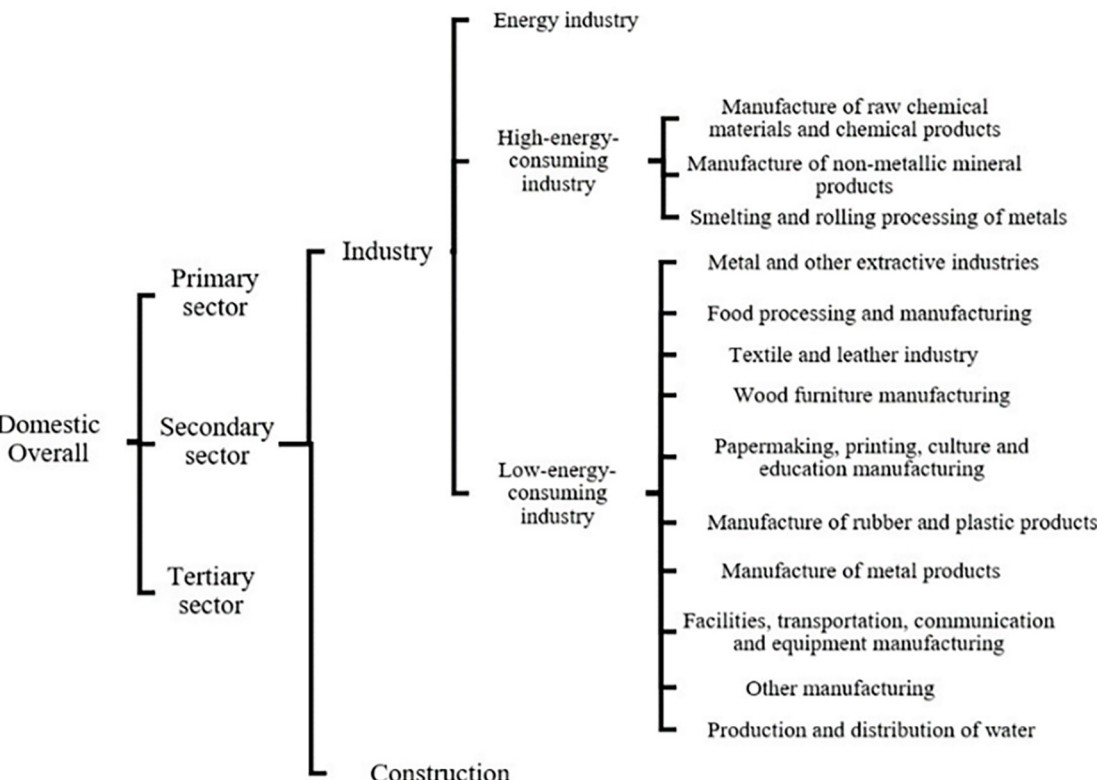

**Fig 1. Hierarchy of the calculation of the industrial energy rebound effect.**

**Table 1. Descriptive statistics of model variables.**

| Variable | Unit | Observed value | Maximum value | Minimum value | Mean value | Standard deviation |
|---|---|---|---|---|---|---|
| Added value | RMB 100 million, price in 2005 | 78 | 480,140.6 | 15,123.2 | 131,137.8 | 117,813.1 |
| Labor | 10,000 people | 78 | 34,872.0 | 1463.9 | 12,724.2 | 11,210.1 |
| Energy | 10,000 tce | 78 | 170,972.0 | 3486.0 | 61,831.0 | 51,812.7 |
| Capital stock | RMB 100 million, price in 2005 | 78 | 322,3821.6 | 5852.4 | 482,617.9 | 646,307.8 |

industry, high-energy-consuming industry, low-energy-consuming industry, construction, and the tertiary sector. During 2005–2017, the average actual added value of the six industries was RMB 13,113.78 billion, and the actual added value of the low-energy-consuming industry in 2017 was RMB 48,014.06 billion, the maximum during the study period; the actual added value of the primary sector in 2005 was RMB 1512.32 billion, the minimum during the study period. The average labor in the six subsectors was 127.242 million. In 2017, the tertiary sector had a labor of 348.72 million, the maximum during the study period; whereas the energy industry in the same year had a labor of 14.639 million, the minimum during the study period. The average energy consumption of the six subsectors was 618.31 million tce. In 2014, the energy consumption of the high-energy-consuming industry was 1709.72 million tce, the maximum during the study period; and the energy consumption of construction in 2005 was 34.86 million tce, the minimum during the study period. The average actual capital stock of the six subsectors was RMB 48,261.79 billion. In 2017, the actual capital stock of the tertiary sector was RMB 322,382.16 billion, the maximum during the study period; and the actual capital stock of construction in 2005 was RMB 585.24 billion, the minimum during the study period.

The descriptive statistics of the subsector variables in this study focus on the fourth level. Table 2 shows the statistics of three high-energy-consuming industries, namely, manufacturers of chemical raw materials and chemical products, manufacturers of nonmetallic mineral products, and smelting and rolling processing of metals and 10 low-energy-consuming industries, incorporating 13 subsectors (namely, metal and other extractive industries, food processing and manufacturing, textile and leather industry, wood furniture manufacturing, papermaking, printing, culture and education manufacturing, manufacturers of rubber and plastic products, manufacturers of metal products, facilities, transportation, communication and equipment manufacturing, other manufacturing, and production and distribution of water). During 2005–2017, the average actual added value of the 13 subsectors was RMB 3386.68 billion, and the actual added value of facilities, transportation, communication, and equipment manufacturing in 2017 was RMB 21,686.05 billion, the maximum during the study period. The actual added value of production and distribution of water in 2005 was RMB 94.11 billion, the minimum during the study period. The average labor in the 13 subsectors was 13.789 million. In 2013, the facilities, transportation, communication, and equipment manufacturing had a labor of 59.256 million, the maximum during the study period; and in 2013, the production and distribution of water had a labor of 758,000, the minimum during the study period. The average energy consumption of the 13 subsectors was 167.191 million tce. In 2014, the energy consumption of the smelting and rolling processing of metals was 975.583 million tce,

**Table 2. Descriptive statistics of subsector variables.**

| Variable | Unit | Observed value | Maximum value | Minimum value | Mean value | Standard deviation |
|---|---|---|---|---|---|---|
| Added value | RMB 100 million, price in 2005 | 169 | 216,860.5 | 941.1 | 33,866.8 | 39,499.8 |
| Labor | 10,000 people | 169 | 5925.6 | 75.8 | 1378.9 | 1640.2 |
| Energy | 10,000 tce | 169 | 97,558.3 | 710.9 | 16,719.1 | 23,038.2 |
| Capital stock | RMB 100 million, price in 2005 | 169 | 679,067.5 | 2309.4 | 82,566.5 | 108,566.2 |

the maximum during the study period; and the energy consumption of the production and distribution of water in 2005 was 7.109 million tce, the minimum during the study period. The average actual capital stock of the 13 subsectors was RMB 8256.65 billion. In 2017, the actual capital stock of facilities, transportation, communication, and equipment manufacturing was RMB 67,906.75 billion, the maximum during the study period; and the actual capital stock of other manufacturing in 2005 was RMB 230.94 billion, the minimum during the study period.

## IV. Model results and analyses

### (I) Calculation results of the growth rate in TFP

Based on the data obtained, the Malmquist index values of China's overall economy, each sector, and various industries within the secondary sector during 2005–2017 are calculated, and the GTFP is obtained from Eq (19). Table 3 shows the results.

Table 3 shows that during 2005–2017, the TFP of the domestic overall and primary, secondary, and tertiary sectors was increasing. Here the generalized technological progress of the domestic overall and secondary sector was prominent, whereas that of the primary and tertiary sectors was low over a long term. During 2005–2017, the average TFP growth of industry was significantly higher than that of construction, with a total growth rate of 89.1%. The GTFP of energy-consuming industries within industry was significantly higher than that of the energy industry, and the total growth rates of the high-energy-consuming industry, low-energy-consuming industry, and energy industry were 55.1%, 64.4%, and 21.8%, respectively. Additionally, the GTFP of construction was less than 0 during 2011–2013, which indicates that technological progress was not realized during this period. The industry GTFP was relatively high during 2005–2007, but experienced a sharp drop during 2007–2009, followed by a gradual increase in the following years. The GTFP of the energy industry was relatively high during 2005–2007, and remained low in all other years. The GTFP of the high-energy-consuming industry was relatively high during 2005–2007 and 2011–2013, and decreased gradually in the other years. The GTFP of the low-energy-consuming industry was relatively low during 2011–2013, and increased gradually in other years.

### (II) Calculation results of the DEI

The DEI of the domestic overall, primary, secondary, and tertiary sectors, and various industries within the secondary sector are obtained from Eq (23). The results are shown in Table 4.

Table 3. The GTFP of China's domestic overall, three sectors, and various industries within the secondary sector during 2005–2017.

| Period | Domestic overall | Primary sector | Secondary sector | #Industry | *Energy industry | *High-energy-consuming industry | *Low-energy-consuming industry | #Construction | Tertiary sector |
|---|---|---|---|---|---|---|---|---|---|
| 2005–2007 | 0.115 | 0.02 | 0.14 | 0.136 | 0.177 | 0.206 | 0.112 | 0.08 | 0.066 |
| 2007–2009 | 0.063 | 0.04 | 0.083 | 0.081 | 0.005 | 0.034 | 0.086 | 0.096 | 0.043 |
| 2009–2011 | 0.062 | 0.002 | 0.093 | 0.099 | 0.005 | 0.02 | 0.097 | 0.01 | 0.028 |
| 2011–2013 | 0.046 | 0.013 | 0.078 | 0.106 | 0.01 | 0.123 | 0.045 | −0.041 | 0.01 |
| 2013–2015 | 0.038 | 0.02 | 0.06 | 0.102 | 0.008 | 0.048 | 0.075 | 0.011 | 0.028 |
| 2015–2017 | 0.034 | 0.005 | 0.072 | 0.15 | 0.006 | 0.036 | 0.105 | 0.003 | 0.018 |
| 2005–2017 | 0.413 | 0.104 | 0.653 | 0.891 | 0.218 | 0.551 | 0.644 | 0.15 | 0.208 |

**Table 4. The DEI of the domestic overall, primary, secondary, and tertiary sectors, and various industries within the secondary sector during 2005–2017.**

| Period | Domestic overall | Primary sector | Secondary sector | #Industry | *Energy industry | *High-energy-consuming industry | *Low-energy-consuming industry | #Construction | Tertiary sector |
|---|---|---|---|---|---|---|---|---|---|
| 2005–2007 | 0.080 | 0.053 | 0.087 | 0.083 | 0.094 | 0.127 | 0.109 | 0.130 | 0.125 |
| 2007–2009 | 0.112 | 0.108 | 0.122 | 0.115 | −0.061 | 0.136 | 0.164 | 0.161 | 0.113 |
| 2009–2011 | 0.053 | 0.015 | 0.090 | 0.093 | 0.003 | 0.111 | 0.145 | 0.028 | 0.020 |
| 2011–2013 | 0.080 | 0.034 | 0.116 | 0.115 | 0.006 | 0.139 | 0.136 | 0.039 | 0.006 |
| 2013–2015 | 0.114 | 0.058 | 0.128 | 0.127 | 0.003 | 0.142 | 0.154 | 0.072 | 0.072 |
| 2015–2017 | 0.094 | 0.016 | 0.111 | 0.114 | −0.097 | 0.084 | 0.121 | 0.007 | 0.043 |
| 2005–2017 | 0.665 | 0.303 | 0.860 | 0.847 | −0.064 | 1.007 | 1.173 | 0.492 | 0.437 |

Table 4 shows that the DEI fluctuates in each period. Throughout the study period, the DEI of the secondary sector remained the highest, and those of the tertiary and primary sectors were relatively low. The DEI of the tertiary sector was particularly low during 2009–2011 and 2011–2013, which is the main reason for the particularly high energy rebound effect of the tertiary sector in these two periods. The domestic overall DEI is relatively high, which is consistent with the implementation of the obligatory targets of energy intensity reduction across China since the Eleventh Five-Year Plan. The DEI of industry is relatively high, yet the DEI of construction is relatively low. The DEIs of both high- and low-energy-consuming industries within industry are high, where the low-energy-consuming industry shows a gradual decreasing trend, and the high-energy-consuming industry shows fluctuations. The DEI of energy industry is very low, with negative values in two time periods, indicating that the energy intensity of the energy industry increased instead of decreasing in these time periods, and that the energy conservation targets failed. The fact that the DEI of the energy industry is very low is the main reason for the high rebound effect of the energy industry.

## (III) The energy rebound effect of China's overall economy and the three sectors

The energy rebound effects of the domestic overall, primary, secondary, and tertiary sectors, and various industries within the secondary sector can be obtained from Eqs (19)–(23). The results are shown in Table 5.

Table 5 shows that the energy rebound effects of the domestic overall, and primary, secondary, and tertiary sectors emerged during 2005–2017, and most of the energy rebound effects are less than 1, with a few above 1, indicating that improvements in energy efficiency still play a positive role in promoting energy conservation in most years, despite falling behind the theoretical expectations.

The energy rebound effect of the domestic overall during 2005–2017 was 53%; it rose during 2009–2011 and declined gradually in other years. The average result of the domestic overall calculated in this study is not too different from the existing literature. The energy rebound effect of the domestic overall calculated by domestic scholars is 30%–80% [17].

The energy rebound effect of the primary sector is relatively low, which was 32% during 2005–2017. This can be explained by the relatively low DEI of the primary sector (Table 4), which leads to relatively low theoretical energy savings. Besides, the results in Table 3 show

**Table 5. Energy rebound effects of China's domestic overall, the three sectors, and various industries within the secondary sector during 2005–2017.**

| Period | Domestic overall | Primary sector | Secondary sector | #Industry | *Energy industry | *High-energy-consuming industry | *Low-energy-consuming industry | #Construction | Tertiary sector |
|---|---|---|---|---|---|---|---|---|---|
| 2005–2007 | 1.12 | 0.35 | 1.23 | 1.26 | 1.44 | 1.07 | 0.78 | 0.45 | 0.40 |
| 2007–2009 | 0.47 | 0.34 | 0.56 | 0.59 | – | 0.34 | 0.32 | 0.46 | 0.31 |
| 2009–2011 | 0.97 | 0.13 | 0.83 | 0.86 | 1.34 | 0.41 | 0.38 | 0.29 | 1.19 |
| 2011–2013 | 0.50 | 0.36 | 0.58 | 0.79 | 1.52 | 0.64 | 0.28 | – | 1.32 |
| 2013–2015 | 0.29 | 0.32 | 0.41 | 0.71 | 2.68 | 0.38 | 0.46 | 0.13 | 0.33 |
| 2015–2017 | 0.32 | 0.29 | 0.58 | 1.17 | – | 0.29 | 0.74 | 0.40 | 0.35 |
| 2005–2017 | 0.53 | 0.32 | 0.65 | 0.87 | – | 0.51 | 0.58 | 0.40 | 0.41 |

that the GTFP of the primary sector is relatively low, indicating a smaller amount of new energy resulting from technological progress. Overall, the impact of slow technological progress is greater than that of small DEI; thereby, its energy rebound effect is much lower than those of other industries.

As the focus of China's energy conservation and consumption reduction policies, the secondary sector features the highest proportion of energy consumption and the strongest energy intensity. Table 5 shows that the rebound effect of the secondary sector is the highest, reaching 65% during 2005–2017. Meanwhile, the rebound effect of the secondary sector fluctuates and decreases gradually during the study period, which is identical to the variation trend of the domestic overall. This indicates that the rebound effect of the domestic overall is affected greatly and directly by the secondary sector.

The rebound effect of the tertiary sector shows strong fluctuations. The energy rebound effect of the tertiary sector during 2005–2017 was 41%, which soared to 119% and 132% during 2009–2011 and 2011–2013, respectively, while remaining below 40% during other periods. Therefore, 2009–2013 was special for the rebound effect of the tertiary sector and a key period to enhance the rebound effect throughout the study period. Table 4 shows that the DEI of the tertiary sector during the above periods was very low, which in turn led to the energy rebound rising above 100%. China has achieved significant industrial restructuring since 2006; the tertiary sector, dominated by service industries, has continued to increase in proportion to the economy, becoming a new engine driving economic development. However, as energy consumption surged during the initial period of rapid development, the growth of added value of industries was significantly lower than that of energy consumption, resulting in a significant slowdown in the DEI of the tertiary sector. This shows that the added value of the tertiary sector during 2009–2013 was not high, and its quality of development needs further improvement. Table 3 shows that the GTFP of the tertiary sector has not been high, and even showed a gradual decline, indicating slow technological progress, which explains the slow growth of the added value of industries.

## (IV) The energy rebound effect of various industries within the secondary sector

Table 5 shows that the rebound effect of industry is relatively high, and the rebound effect of construction still shows a gradual decline despite rising slightly during 2015–2017. The

rebound effect of industry during 2005–2017 reached 87%, and the rebound effects of high- and low-energy-consuming industries were 51% and 58%, respectively, which were far below the total rebound effect of industry. It can be inferred that the high rebound effect of industry is mainly because of the energy industry. A detailed analysis of the energy industry and high- and low-energy-consuming industries is as follows.

1. (1)The energy rebound effect of the energy industry

Tables 3 and 4 show that the GTFP and DEI of the energy industry are relatively low. Overall, the impact of the DEI is greater than that of the GTFP, which is manifested by its all-time-high energy rebound effect. As mentioned earlier, the decline in energy intensity reflects not only the level of technological development of the industry but is also the result of the industry's products being affected by supply and demand relationships or pricing control. If the price of primary and secondary energy produced by the energy industry is relatively low, the added value will be low, too, and the energy intensity will remain high. In other words, the energy price parameter is an important reason for the low DEI of the energy intensity.

Taking electricity and gas production and supply as an example, which are the pillar sectors in the energy industry, although the proportion of renewable energy power generation in China has been rising in recent years, coal consumption for national coal power generation dropped from 359 to 301 gce/kWh (Calculations are based on the Energy Balance Sheets in 2005 and 2017). However, the overall energy efficiency improvements of the energy industry are not equal to those of the high-energy-consuming industry. As shown in Figs 2 and 3, the energy industry's share of total GDP dropped from 8.96% to 4.39%, and its share of added value of industry dropped from 22% to 13%, indicating that the increase of the added value of the energy industry during the study period was far below that of the overall added value of industry.

The low DEI of energy industry is mainly due to the long-term low pricing of energy prices. Fossil energy subsidies artificially depress the price of fossil energy, thus lowering the total energy price and increasing the energy intensity. Fossil energy subsidies have been going on for a long time in China, even after past decades' reforms aiming at establishing a market-oriented economy, the price reform is still stagnant in energy market. This is because the energy industry is located in the upstream of the macro-economic chain of production and plays an important role in the economy. Therefore, the Chinese government tends to keep intervening in the energy market, which leads to distorted energy prices [36–38].

1. (2) The energy rebound effect of the energy-consuming industry

As regards the energy-consuming industry, the energy conservation effect is more significant. The rebound effects of high- and low-energy-consuming industries during 2005–2017 were 51% and 58%, respectively, indicating that the energy-consuming industry achieved nearly 50% of the energy conservation targets.

During the study period, the rebound effect of the high-energy-consuming industry showed a gradual decline with fluctuations, whereas that of the low-energy-consuming industry showed a gradual increase. Meanwhile, the proportion of the high-energy-consuming industry in added value of industry has stabilized at 22% after 2007, whereas that of the low-energy-consuming industry rose from 58% to 64%; the proportion of the high-energy-consuming industry in total GDP dropped from 8.59% in 2005 to 7.34% in 2017, and that of the low-energy-consuming industry dropped from 24.1% to 21.34% (Figs 2 and 3).

Based on the data collected and processed in this study, the Malmquist index of each sub-sector within the energy-consuming industry during 2005–2017 can be estimated, and the energy rebound effect can be calculated from Eqs (9)–(13). Table 6 shows that the rebound

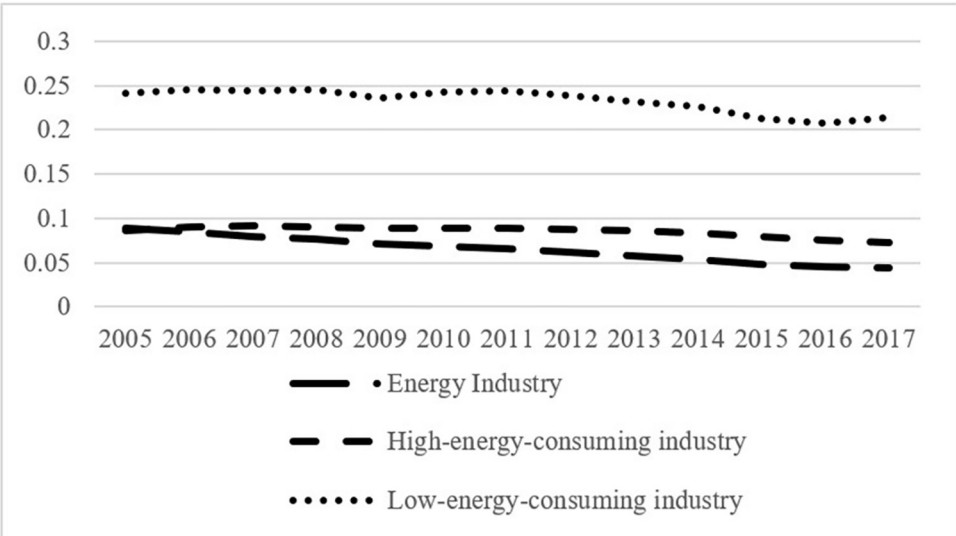

**Fig 2. Changes in the internal industrial structure of industry during 2005–2017 (proportion of the total GDP).**

effects of the manufacture of chemical raw materials and chemical products, manufacture of nonmetallic mineral products, and smelting and rolling processing of metals in the high-energy-consuming industry show a gradual decline. The rebound effects of traditional manufacturing industries, such as metal and other extractive industries, food processing, and manufacturing in the low-energy-consuming industry are relatively low, which is consistent with the low technological progress rate; the rebound effects of light industries, such as textile and leather industry, wood furniture manufacturing, papermaking, printing, culture, and education, were about 50% during the study period, showing a gradual increase; and the rebound effects of facilities, transportation, communication, and equipment manufacturing, including

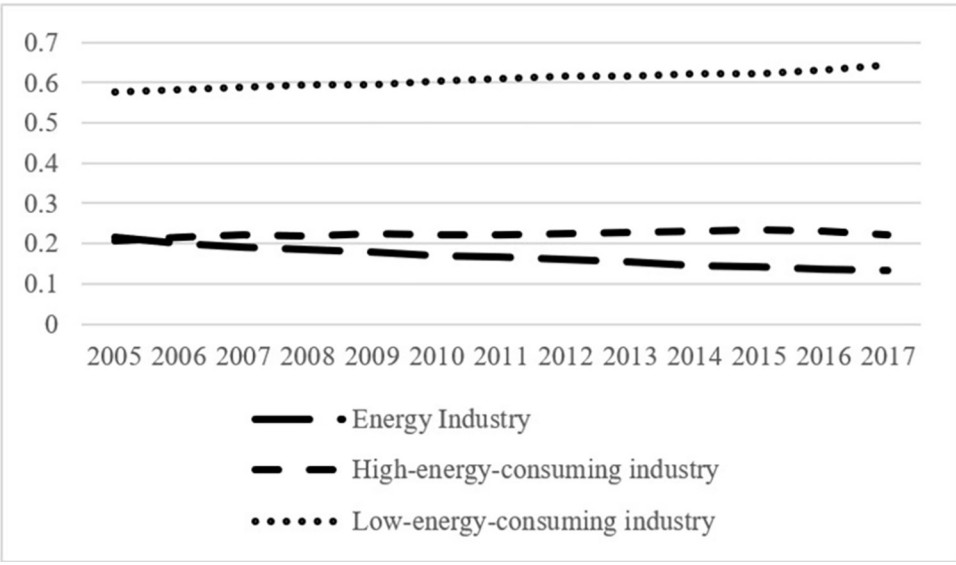

**Fig 3. Changes in the internal industrial structure of industry during 2005–2017 (proportion of added value of industry).**

**Table 6. Energy rebound effects of subsectors within the energy-consuming industry during 2005–2017.**

| | 2005–2007 | 2007–2009 | 2009–2011 | 2011–2013 | 2013–2015 | 2015–2017 | 2005–2017 |
|---|---|---|---|---|---|---|---|
| Manufacture of raw chemical materials and chemical products | 0.55 | 0.31 | 0.41 | 0.36 | 0.30 | 0.23 | 0.42 |
| Manufacture of nonmetallic mineral products | 0.73 | 0.36 | 0.56 | - | 0.45 | 0.28 | 0.39 |
| Smelting and rolling processing of metals | - | 0.34 | 0.54 | 0.93 | 0.43 | 0.37 | 0.71 |
| Metal and other extractive industries[a] | 0.10 | 0.01 | 0.02 | 0.01 | 0.04 | 0.06 | 0.03 |
| Food processing and manufacturing | 0.07 | 0.02 | 0.50 | 0.08 | 0.09 | 0.26 | 0.07 |
| Textile and leather industry | 0.83 | 0.03 | 0.26 | 0.85 | 0.89 | 0.91 | 0.47 |
| Wood furniture manufacturing | 0.76 | 0.11 | 0.19 | 0.49 | 0.88 | 0.87 | 0.52 |
| Papermaking, printing, culture, and education | 0.58 | 0.29 | 0.25 | 0.53 | 0.89 | 0.89 | 0.51 |
| Manufacture of rubber and plastic products | 0.39 | - | - | 0.81 | 0.40 | 0.44 | - |
| Manufacture of metal products | 0.23 | 0.01 | 0.29 | - | 0.70 | 0.88 | 0.26 |
| Facilities, transportation, communication, and equipment manufacturing | 0.79 | 0.84 | 0.77 | 0.85 | 0.86 | 0.83 | 0.84 |
| Other manufacturing | 0.25 | - | 0.01 | 0.83 | 0.13 | 0.04 | 0.01 |
| Production and distribution of water | 0.55 | 0.14 | 1.46 | 0.60 | 0.12 | 0.21 | 0.50 |

[a]Metal and other extractive industries include metal and other extractive industries, and the smelting and rolling processing of metals (including ferrous and nonferrous metals).

The "-" in the table represents a meaningless result caused by the recovery of energy intensity or negative GTFP.

high-tech industries, high-end equipment manufacturing, and strategic emerging industries are relatively high.

Hu [21] conducted a simulation analysis of the CGE model based on the input–output table in 2007, and believed that improving the energy efficiency of the high-energy-consuming industry will lead to a stronger rebound effect. However, the conclusions of this study indicate that despite the improvement in energy efficiency of the high-energy-consuming industry during the study period, its energy rebound effect shows a gradual decline. Meanwhile, the industrial structure of the domestic overall does not shift toward the high-energy-consuming industry. In this study, we believe that various practical policy measures aimed at adjusting the industrial structure offset the self-reinforcing feedback effect within the economic system.

Since 2006, China has successively introduced a series of energy conservation and emission reduction and industrial restructuring policy measures to curb the blind expansion of energy-intensive and pollution-intensive industries. The energy rebound effect of the high-energy-consuming industry was higher than that of the low-energy-consuming industry during 2005–2007 and 2011–2013 (Table 5), because its GTFP is much higher than that of the low-energy-consuming industry (Table 3). These two time periods are 2006 and 2011. Before the Eleventh Five-Year Plan (2006–2010), China's heavy chemical industry experienced rapid development. In the mid–late Eleventh Five-Year Plan, to achieve the goal of a 20% reduction in energy intensity, the development of the high-energy-consuming industry was restricted across the country, but this momentum rebounded in the first few years of the Twelfth Five-Year Plan. This indicates that China's efforts at industrial restructuring cannot be achieved immediately.

During the study period, the overall rebound effect of the low-energy-consuming industry was higher than that of the high-energy-consuming industry, and the former showed a gradual increase. Meanwhile, the high energy rebound effect of the low-energy-consuming industry did not apply to the domestic overall in the same way. Table 5 shows that the energy rebound effects of both the industry and domestic overall reduce gradually. This confirms Feng's /viewpoint [23] that when the energy efficiency of the low-energy-consuming industry improves, on the one hand, the energy cost per unit of yield reduces, and the yield level increases through

the income effect, which increases the total demand for energy of the industry and domestic overall; thus, an energy rebound effect is produced. On the other hand, through the linkage of the economic structure and factor substitution effect, the flow of energy factor to the low-energy-consuming industry increases, whereas that to the high-energy-consuming industry decreases; thereby, the energy efficiency level of the domestic overall increases. As the low-energy-consuming industry is composed mainly of new manufacturing and light industries, its products are used mainly as final products for other industrial sectors and consumers, and its high rebound effect will not spread to the domestic overall. Figs 2 and 3 show that although the proportion of the low-energy-consuming industry in added value of industry increases gradually, its proportion in the total GDP declines gradually, along with the decreasing proportion of industry in the total GDP, which explains why the high energy rebound effect of the high-energy-consuming industry does not lead to a high rebound effect of the domestic overall.

The impact resulting from improvements in the low-energy-consuming industry efficiency on the domestic overall energy efficiency can be explained from the primary and secondary energy perspectives. As the low-energy-consuming industry features relatively intensive use of secondary energy, its energy efficiency improvements reduce the domestic overall consumption of secondary energy, and indirectly reduce the conversion of primary energy consumption in the secondary energy. In addition, the promotion of the low-energy-consuming industry driven by energy efficiency improvements will shift the economic structure toward the low-energy-consuming industry, which contributes to reducing the overall energy intensity. Improvements in the efficiency of secondary energy use has a greater effect on the economy, and the rebound effect that comes along is generally less than that by improving the efficiency of primary energy use.

Currently, the energy rebound effects of the high-energy-consuming industry and domestic overall are declining constantly, whereas the energy rebound effect of the low-energy-consuming industry is increasing steadily. This reflects that the high energy rebound effect of the high-energy-consuming industry and domestic overall has been controlled in the wake of China's industrial restructuring policy since the Eleventh Five-Year Plan.

In summary, although the energy efficiency improvements in the low-energy-consuming industry produce rebound effects, they optimize the industry's internal industrial structure and improve domestic overall energy efficiency, while not increasing the energy rebound effect of the domestic overall. Therefore, China's long-term energy efficiency policy that focuses on the high-energy-consuming industry should be shifted toward the low-energy-consuming industry as required. As the low-energy-consuming industry uses mainly secondary energy, it effectively increases its energy cost by increasing the electricity price, which further promotes technological progress, optimizes industrial structure, and improves domestic overall energy efficiency. Liu and Lin [39] simulated the macroeconomic impact of electricity price changes by establishing a dynamic CGE model and found that the same law, that is, the effect of electricity pricing increases, is beneficial to China's industrial restructuring through medium- and long-term accumulation.

## V. Conclusion and suggestions

In conclusion, this study established an energy rebound effect model based on the generalized technological progress; calculated the energy rebound effects of the domestic overall, primary, secondary, and tertiary sectors, and various industries within the secondary sector during 2005–2017; and analyzed the calculation results from the two principal influence parameters, GTFP and DEI, as well as China's relevant policies during the study period. The study shows

that the energy rebound effect of the secondary sector during the study period had a great impact on that of the domestic overall; the energy rebound effects of the domestic overall, industry, high- and low-energy-consuming industries are 53%, 87%, 51%, and 58%, respectively. In some years, the energy intensity of the energy industry will increase instead of decrease, and the rebound effects are above 100% in the remaining years; the energy rebound effects of the domestic overall and high-energy-consuming industry show a downward trend year by year, whereas those of the low-energy-consuming industry show an upward trend year by year. The industrial restructuring policy from the Eleventh Five-Year Plan period has effectively reduced the energy rebound effect of the high-energy-consuming industry; meanwhile, the low-energy-consuming industry with an ever-increasing energy rebound effect has not raised the energy rebound effect of the domestic overall; the energy industry with the high energy rebound effect does not have energy efficiency improvement with economic significance, which is related to the long-term low pricing in electricity; and the high energy rebound effect of the tertiary sector in some years is attributed to the low added value of the industry.

On the other hand, the energy rebound effect can promote welfare through several mechanisms [40]. If the cost of welfare improvement is the negative externality caused by excessive energy consumption, technological progress is a double-edged sword. Although this paper does not focus on the welfare effect, but the technological progress induced economic growth, which is the key to determine the energy rebounded in the definition of RE, could represent the welfare brought by the energy efficiency improvement. The empirical results show that all REs are less than 1, indicating that energy efficiency improvement has played a positive role in all sectors and industries, but different industries have different effects on the RE of the domestic overall. The energy rebound cannot be avoided, but the policy can intervene to optimize the trade-off between the technological progress of the domestic overall and its RE.

The above conclusions cast light on certain policies. First, although technological progress results in rebound effects, it is still an effective measure of energy conservation and emission reduction. Second, the optimization and upgrading of industrial structure is a significant approach for energy conservation and consumption reduction; in particular, it curbs the rebound of the high-energy-consuming industry tangibly. Third, the industrial heterogeneity of energy rebound determines that strategies aimed at reducing the energy rebound effect should be adapted to local conditions, and the long-term energy efficiency policy that focuses on the high-energy-consuming industry should be shifted toward the low-energy-consuming industry. Since the high-energy-consuming industries mainly use primary fossil fuel such as coal and oil, while the low-energy-consuming industries mainly use secondary energy such as electricity, the energy efficiency of the low-energy-consuming industry can be encouraged by improving the energy efficiency of the electrical equipment rather than coal equipment. In addition, we can also improve the energy cost of primary fossil fuel by formulating reasonable environmental regulation policies, further promote the flow of energy factor from high-energy-consuming industries to low-energy-consuming industries, and realize the low energy rebound effect of the domestic overall. Fourth, the pricing mechanism, especially the pricing in electricity, can effectively reduce the energy rebound effect of the energy industry, further improve the energy efficiency of the low-energy-consuming industry, and promote industrial structure optimization within the industry. Finally, only sustained efforts in promoting the technological progress of the tertiary sector and increasing its added value can effectively improve the energy efficiency of the industry with economic significance.

In recent years, the research on energy rebound effect has gradually increased. Although the energy rebound effect has not been taken into account in policy making and evaluation, however the government's actions in the industrial structure adjustment just offset the energy rebound effect. In the future, with the deepening of the research on energy rebound effect, if

the energy rebound effect of high-energy-consuming industry is proved to exist and the government does not take it into account when making policies, the energy rebound effect may become more and more serious. In addition, the current energy price can't reflect the real cost and the relationship between market supply and demand, which hinders the research on the rebound effect of energy. It is expected that the distortion of energy price can be alleviated in the future, or better research methods and technologies can be found to avoid this problem.

## Author Contributions

**Conceptualization:** Guangqing Xu.

**Data curation:** Xiaoyu Chen.

**Formal analysis:** Guangqing Xu.

**Funding acquisition:** Guangqing Xu.

**Software:** Danze Huang.

**Writing – original draft:** Guangqing Xu, Mengyao Chen.

**Writing – review & editing:** Guangqing Xu, Mengyao Chen.

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
