## [Decision Letter · Decision Letter 0]

5 Dec 2022

PONE-D-22-02801Study on energy rebound effects of China’s industriesPLOS ONE

Dear Dr. Chen,

Thank you for submitting your manuscript to PLOS ONE. After careful consideration, we feel that it has merit but does not fully meet PLOS ONE’s publication criteria as it currently stands. Therefore, we invite you to submit a revised version of the manuscript that addresses the points raised during the review process.

 Please address each reviewer comment carefully. In particular, you will find that the reviewers have some concerns about the motivation, the data and methods, and the discussion of the results. Please make sure that you provide a point-by-point response to the reviewers' comments. 

We look forward to receiving your revised manuscript.

Kind regards,

Yueming Qiu

Academic Editor

PLOS ONE

Journal Requirements:

“This work was supported by fund for building world-class universities (disciplines) of Renmin University of China (Rroject No. 2022). The funders had no role in study design, data collection and analysis, decision to publish, or preparation of the manuscript.”

Reviewers' comments:

Reviewer's Responses to Questions

**Comments to the Author**

1. Is the manuscript technically sound, and do the data support the conclusions?

Reviewer #1: Yes

Reviewer #2: Partly

2. Has the statistical analysis been performed appropriately and rigorously? 

Reviewer #1: Yes

Reviewer #2: Yes

3. Have the authors made all data underlying the findings in their manuscript fully available?

Reviewer #1: Yes

Reviewer #2: Yes

4. Is the manuscript presented in an intelligible fashion and written in standard English?

Reviewer #1: Yes

Reviewer #2: Yes

5. Review Comments to the Author

Reviewer #1: Energy rebound effects exist popularly and play important roles in reality. Therefore, this article addresses an interesting issue. Conclusions are interesting and robust. This is a well organized paper and I recommend to publish it with modification.

Minor remarks

1. Authors should explain the reason to adopt data between 2005 -2017. There is new data about this research.

2. More recent papers about rebound effects should be addressed.

[1]Miao, Z., & Chen, X. (2022). Combining parametric and non-parametric approach, variable & source-specific productivity changes and rebound effect of energy & environment. Technological Forecasting and Social Change, 175, 121368.

[2]Wang, C., & Nie, P. Y. (2018). How rebound effects of efficiency improvement and price jump of energy influence energy consumption?. Journal of Cleaner Production, 202, 497-503.

Reviewer #2: This paper utilized the Chinese statistic yearbook data to estimate the energy consumption rebound effects of the domestic overall and different industries. The methodology, evidence, and results are clear. The manuscript is also written in a structured and clear manner. I appreciate the efforts that the authors spent on putting this together. However, I still have several concerns and this paper could be improved further. My detailed comments and concerns are as follows:

Major comments:

• The motivation of this paper is not discussed in a clear way in the introduction. This paper mentions that many studies have predicted carbon emissions of the economy in China under different scenarios but these studies have not taken into account the impact of energy rebound effect when predicting carbon emissions (lines 20-25). I do not fully understand this argument as a reader. For the existing literature, how did they predict carbon emissions and how did they omit the energy rebound effects? From my perspective, if researchers predict future GDP growth and future energy intensity (determined by technological progress), they can naturally obtain future energy consumption and related emissions, in which they have taken the energy rebound effect into account. Thus, I think the authors must clarify this very first argument. In addition, the authors should also discuss how should we use the results of this paper to better predict carbon emissions (at least in the discussion section)? How should we apply the finding of this paper to future studies?

• I do not fully understand the theoretical model derivation. I have three concerns: First, following the fundamental definition of energy rebound effects, this paper should show the equations/functions of actual savings and theoretical savings. The authors have shown the function form of theoretical savings but not the actual savings. For me, it is not clear what the function of actual savings looks like in this paper. Second, from my understanding, theoretical saving is based on an assumption that demand does not change. In this meaning, the base year in equation (3) should be Yt instead of Yt+1. Third, this paper uses the ratio of GTFP (growth of TFP) to the growth of economy to measure the contribution of technological progress to economic growth (line 160, equation 11). I am not sure whether this measurement is reasonable or not. The authors need to clarify this argument. What are the assumptions and limitations?

• This paper is focused on carbon emission reductions (addressing negative externalities) when discussing the impact of energy rebound effect. This may not be the whole picture if we consider the comprehensive welfare effects of energy rebound. On the other hand, “the macroeconomic price effect of an energy efficiency improvement arises from reaching equilibria in markets, which improves welfare. Sectoral reallocation leads to more efficient production in an economy, improving welfare. If the energy efficiency improvement induces innovation, this would also improve welfare” (Gillingham, Rapson, & Wagner, 2016). I suggest that this paper should discuss the energy rebound effect comprehensively.

Minor points:

• Table 2 can also show the descriptive statistics by different sectors or industries.

• On line 438, this paper mentions “the long-term energy efficiency policy that focuses on the high-energy-consuming industry should be shifted toward the low energy-consuming industry.” It could be much better to lay out specific policy suggestions for the low-energy-consuming industry.

6. PLOS authors have the option to publish the peer review history of their article (what does this mean?). If published, this will include your full peer review and any attached files.

Reviewer #1: No

Reviewer #2: No

---

## [Author Response · Author response to Decision Letter 0]

22 Jan 2023

Some formatting errors occurred, please refer to the file “Response to Reviewers”.

Thanks to the sharp and highly constructive comments and hard work of the reviewers and the efficient work of the academic editor Dr.Qiu, the authors has carefully revised the manuscript according to the comments of the reviewers. The following is a point-by-point response to the comments.

Reviewer #1：

1. Authors should explain the reason to adopt data between 2005 -2017. There is new data about this research.

Responses：

The reason for choosing 2005 as the starting year is that the statistical caliber of China’s industrial data before 2005 is slightly different from that of 2005 and after. The end year of 2017 is mainly due to the limited availability of data when we first finished our work at the middle of 2020, .when only the data of industrial sub-sectors of the year 2017 and before were available. It took more than one year to modify and translate into English, and the paper was submitted in the end of 2021.

2. More recent papers about rebound effects should be addressed.[1]Miao, Z., & Chen, X. (2022). Combining parametric and non-parametric approach, variable & source-specific productivity changes and rebound effect of energy & environment. Technological Forecasting and Social Change, 175, 121368.[2]Wang, C., & Nie, P. Y. (2018). How rebound effects of efficiency improvement and price jump of energy influence energy consumption?. Journal of Cleaner Production, 202, 497-503.

Responses:

Thanks for the reviewer's suggestions, we have added the introduction of these two literatures to the literature review, which is as follows:

“In addition, some scholars studied the influencing factors of rebound effect. Wang and Nie (2018) established a Cournot competition model and studied the influence of competition degree and price fluctuation on energy rebound effect. Miao and Chen (2022) developed a non-parametric framework variable-specifically and source-specifically to investigate the impact mechanism of endogenous Total Factor Productivity (TFP) on rebound effect (RE).”

Reviewer #2：

1. This paper mentions that many studies have predicted carbon emissions of the economy in China under different scenarios but these studies have not taken into account the impact of energy rebound effect when predicting carbon emissions (lines 20-25). I do not fully understand this argument as a reader. For the existing literature, how did they predict carbon emissions and how did they omit the energy rebound effects? From my perspective, if researchers predict future GDP growth and future energy intensity (determined by technological progress), they can naturally obtain future energy consumption and related emissions, in which they have taken the energy rebound effect into account. Thus, I think the authors must clarify this very first argument. In addition, the authors should also discuss how should we use the results of this paper to better predict carbon emissions (at least in the discussion section)? How should we apply the finding of this paper to future studies?

Responses：

Thanks for the reviewer's sharp comments. In fact the rebound effect is not specifically included in the analysis of many carbon emission projections. With the further decrease of energy intensity in the future, the impact of rebound effect may still exit. The rebound effect should be taken into account by adding a discount rate into the prediction of energy intensity related energy reduction, which may be represented as:

Actual Energy Savings=Theoretical Energy Savings *(1-RE)

Where, RE means rebound effect.

We revised the theoretical model derivation part(see the responses of comments 2) and addressed some discussion in term of how to taking into account the energy rebound effect in the carbon emission predictions, which is as follows:

“When predicting carbon emissions based on energy consumption, the energy rebound effect should be taken into account by adding a discount rate into the energy efficiency improvement related energy reduction, which may be represented as:Actual Energy Savings=Theoretical Energy Savings *(1-RE), Where, RE means rebound effect.”

2. I do not fully understand the theoretical model derivation. I have three concerns: 

First, following the fundamental definition of energy rebound effects, this paper should show the equations/functions of actual savings and theoretical savings. The authors have shown the function form of theoretical savings but not the actual savings. For me, it is not clear what the function of actual savings looks like in this paper.

Second, from my understanding, theoretical saving is based on an assumption that demand does not change. In this meaning, the base year in equation (3) should be Yt instead of Yt+1.

Third, this paper uses the ratio of GTFP (growth of TFP) to the growth of economy to measure the contribution of technological progress to economic growth (line 160, equation 11). I am not sure whether this measurement is reasonable or not. The authors need to clarify this argument. What are the assumptions and limitations?

Responses：

Thanks for the reviewer's sharp and helpful comments. We rewrite the whole theoretical derivation part in the manuscript, the following is the point-by-point responses, the detailed derivation is in the revised manuscript.

To the First Comment, the Actual Energy Savings is really very important both in the concept and in the practical meaning, but it can’t be measured directly. The relationship of Actual Energy Savings and the formula of RE is as the following:

“Saunders (1992) and Berkout et al. (2000) defined the energy rebound effect as the ratio of the difference between the theoretical and actual energy savings to the theoretical energy savings after energy efficiency improvements, such difference called the energy rebounded.

RE=(Energy Rebounded)/(Theoretical Energy Savings) (1)

Where, 

Energy Rebounded=Theoretical Energy Savings-Actual Energy Savings (2)

Thereby, 

RE=1-(Actual Energy Savings)/( Theoretical Energy Savings) (3)

Actual Energy Savings=Theoretical Energy Savings*(1-RE) (4)”

To the Second Comment, in our definition and derivation, theoretical savings is not based on an assumption that demand does not change. It is clear that the energy consumption in year t is E_t=Y_t∙EI_t, and the energy consumption in year t+1 is E_(t+1)=Y_(t+1)∙EI_(t+1). The definition of theoretical energy savings can take two forms. Assuming the yield is not growing and the theoretical energy savings is Y_t∙(EI_t-EI_(t+1) ), this definition is very reasonable. However, considering that yield is not only caused by technological progress, but also caused by factors input, scale return, etc., so the definition of theoretical energy savings should consider the yield growth. On this assumption, the theoretical energy savings is Y_(t+1)∙(EI_t-EI_(t+1) ). To sum up, we believe that theoretical energy savings could be defined in these two ways, and the first literature (Zhou and Lin, 2007) that measured the energy rebound effect at the macro level in China defined the theoretical energy savings as Y_(t+1)∙(EI_t-EI_(t+1) ). So do we.

“With reference to Zhou and Lin (2007), suppose Y_t refers to the actual yield, E_t refers to the energy consumption, and EI_t refers to the energy intensity; then, the formula of energy consumption in year t is expressed as:E_t=Y_t∙EI_t. In year t+1, the economic yield is Y_(t+1), the energy intensity decreased from E_t to EI_(t+1) because of technological progress, then, the formula of energy consumption in year t+1 is expressed as:E_(t+1)=Y_(t+1)∙EI_(t+1). Suppose there is no technological progress, the energy consumption in year t+1 will be: E_(t+1)=Y_(t+1)×EI_t. Thereby we get the energy savings because of energy intensity decrease: Y_(t+1)∙(EI_t-EI_(t+1) ). We call it Theoretical Energy Savings, it means the anticipated energy savings from technological progress.

Theoretical Energy Savings=Y_(t+1)∙(EI_t-EI_(t+1) ) (5)”

To the Third Comment, indeed, the formula of the contribution of technological progress to economic growth σ_(t+1) was too simple to understand for readers, so we add a new part which named “estimation of σ_(t+1)”. The revision is as the following:

“Suppose the production function including three input factors of capital K, labor L and energy E as follows:

Y=TFP(t)∙F(K,L,E) (8)

where Y represents economic yield, TFP(t) is total factor productivity, which represents generalized technological progress that change over time, K represents capital, L represents labor, and E represents energy, that is, the total yield of the economy is determined by these four factors. Then, take the derivative of t on both sides of the equation to get:

dY/dt=dTFP/dt∙F(K,L,E)+TFP∙dF/dK∙dK/dt+TFP∙dF/dL∙dL/dt+TFP∙dF/dE∙dE/dt (9)

Divide both sides of the equation by Y to get:

dY/(dt∙Y)=dTFP/(dt∙TFP)+TFP∙dF/dK∙dK/(dt∙Y)+TFP∙dF/dL∙dL/(dt∙Y)+TFP∙dF/dE∙dE/(dt∙Y) (10)

Suppose that in a perfectly competitive market, the price of an input factor is equal to its marginal product, so the following equation exits, where r, w, p represent the price of capital, labor and energy respectively, and α, β, γ represent the output elasticity of capital, labor and energy respectively. 

{█(r=TFP∙dF/dK=α∙Y/K@w=TFP∙dF/dL=β∙Y/L@p=TFP∙dF/dE=γ∙Y/E)┤ (11)

Thus:

dY/(dt∙Y)=dTFP/(dt∙TFP)+α∙dK/(dt∙K)+β∙dL/(dt∙L)+γ∙dE/(dt∙E) (12)

Equation (12) can be written as:

∆Y/Y=∆TFP/TFP+α∙∆K/K+β∙∆L/L+γ∙∆E/E (13)

Then, the growth rate of economic yield can be obtained as follows:

GY=GTFP-α∙GK-β∙GL-γ∙GE (14)

Where, GY, GTFP, GK, GL and GE respectively represent the growth rate of Y, TFP, K, L and E relative to the previous year. According to equation (14), the contribution rate of the growth rate of TFP and the growth rate of each input factor to the growth rate of Y can be estimated respectively. Among them, the contribution rate of the growth rate of TFP to the growth rate of Y is as follows:

σ_(t+1)=〖GTFP〗_(t+1)/〖GY〗_(t+1) (15)”

3. This paper is focused on carbon emission reductions (addressing negative externalities) when discussing the impact of energy rebound effect. This may not be the whole picture if we consider the comprehensive welfare effects of energy rebound. On the other hand, “the macroeconomic price effect of an energy efficiency improvement arises from reaching equilibria in markets, which improves welfare. Sectoral reallocation leads to more efficient production in an economy, improving welfare. If the energy efficiency improvement induces innovation, this would also improve welfare” (Gillingham, Rapson, & Wagner, 2016). I suggest that this paper should discuss the energy rebound effect comprehensively.

Responses:

The comment is quite constructive and right that there should be a whole picture of the rebound effect. As mentioned in the previous response, the improvement in energy efficiency represents technological progress, which promotes economic growth. Our model actual takes economic growth into account, so we consider the contribution of technological progress to economic growth as the key variable to measure the energy rebounded. The technological progress induced economic growth could represent the welfare brought by the energy efficiency improvement. We may focus more on the welfare effect in the next paper. We add some discuss about this as the following and adopt the above literature, thanks for the comment.

“On the other hand, the energy rebound effect can promote welfare through several mechanisms (Gillingham, Rapson, & Wagner, 2016). If the cost of welfare improvement is the negative externality caused by excessive energy consumption, technological progress is a double-edged sword. Although this paper does not focus on the welfare effect, but the technological progress induced economic growth, which is the key to determine the energy rebounded in the definition of RE, could represent the welfare brought by the energy efficiency improvement. The empirical results show that all REs are less than 1, indicating that energy efficiency improvement has played a positive role in all sectors and industries, but different industries have different effects on the RE of the domestic overall. The energy rebound cannot be avoided, but the policy can intervene to optimize the trade-off between the technological progress of the domestic overall and its RE.” 

 4. Table 2 can also show the descriptive statistics by different sectors or industries.

Responses：

Thanks for the reviewer's suggestions. We have shown the descriptive statistics. Table 1 shows the descriptive statistics of six categories of industries: primary industry, tertiary industry, construction industry, energy industry, high energy consumption industry and low energy consumption industry. Table 2 shows the descriptive statistics of 3 high energy consumption industries and 10 low energy consumption industries.

5. On line 438, this paper mentions “the long-term energy efficiency policy that focuses on the high-energy-consuming industry should be shifted toward the low energy-consuming industry.” It could be much better to lay out specific policy suggestions for the low-energy-consuming industry.

Responses：

Thanks for the reviewer's very constructive comments and suggestions. We have added specific policy suggestions for the low-energy-consuming industry to the conclusions and recommendations section of the manuscript.

“Since the high-energy-consuming industries mainly use primary fossil fuel such as coal and oil, while the low-energy-consuming industries mainly use secondary energy such as electricity, the energy efficiency of the low-energy-consuming industry can be encouraged by improving the energy efficiency of the electrical equipment rather than coal equipment. In addition, we can also improve the energy cost of primary fossil fuel by formulating reasonable environmental regulation policies, further promote the flow of energy factor from high-energy-consuming industries to low-energy-consuming industries, and realize the low energy rebound effect of the domestic overall.”

The authors

January 13th,2023

---

## [Decision Letter · Decision Letter 1]

13 Feb 2023

Study on energy rebound effects of China’s industries

PONE-D-22-02801R1

Dear Dr. Chen,

We’re pleased to inform you that your manuscript has been judged scientifically suitable for publication and will be formally accepted for publication once it meets all outstanding technical requirements.

Kind regards,

Yueming Qiu

Academic Editor

PLOS ONE

Additional Editor Comments (optional):

Reviewers' comments:

Reviewer's Responses to Questions

**Comments to the Author**

1. If the authors have adequately addressed your comments raised in a previous round of review and you feel that this manuscript is now acceptable for publication, you may indicate that here to bypass the “Comments to the Author” section, enter your conflict of interest statement in the “Confidential to Editor” section, and submit your "Accept" recommendation.

Reviewer #1: All comments have been addressed

Reviewer #2: All comments have been addressed

2. Is the manuscript technically sound, and do the data support the conclusions?

Reviewer #1: Yes

Reviewer #2: Yes

3. Has the statistical analysis been performed appropriately and rigorously? 

Reviewer #1: Yes

Reviewer #2: Yes

4. Have the authors made all data underlying the findings in their manuscript fully available?

Reviewer #1: Yes

Reviewer #2: Yes

5. Is the manuscript presented in an intelligible fashion and written in standard English?

Reviewer #1: Yes

Reviewer #2: Yes

6. Review Comments to the Author

Reviewer #1: (No Response)

Reviewer #2: Thanks a lot to the authors to address all my comments. The manuscript quality has been further improved. I have no more comments.

7. PLOS authors have the option to publish the peer review history of their article (what does this mean?). If published, this will include your full peer review and any attached files.

Reviewer #1: No

Reviewer #2: No

---

## [Editor Report · Acceptance letter]

20 Feb 2023

PONE-D-22-02801R1 

Study on energy rebound effects of China’s industries 

Dear Dr. Chen:

I'm pleased to inform you that your manuscript has been deemed suitable for publication in PLOS ONE. Congratulations! Your manuscript is now with our production department. 

Kind regards, 

on behalf of

Dr. Yueming Qiu 

Academic Editor

PLOS ONE